# Stochastic Inoculum, Biotic Filtering and Species-Specific Seed Transmission Shape the Rare Microbiome of Plants

**DOI:** 10.3390/life12091372

**Published:** 2022-09-02

**Authors:** David Johnston-Monje, Janneth P. Gutiérrez, Luis Augusto Becerra Lopez-Lavalle

**Affiliations:** 1Max Planck Tandem Group in Plant Microbial Ecology, Universidad del Valle, Cali 76001, Colombia; 2International Center for Tropical Agriculture (CIAT), Cali 763537, Colombia; 3Department of Plant Microbe Interactions, Max Planck Institute for Plant Breeding Research, 50829 Cologne, Germany

**Keywords:** rhizosphere, phyllosphere, endophyte, plant microbiome, plant mycobiome, rare microbiome, soil microbiology, seed microbiome, vertical transmission

## Abstract

A plant’s health and productivity is influenced by its associated microbes. Although the common/core microbiome is often thought to be the most influential, significant numbers of rare or uncommon microbes (e.g., specialized endosymbionts) may also play an important role in the health and productivity of certain plants in certain environments. To help identify rare/specialized bacteria and fungi in the most important angiosperm plants, we contrasted microbiomes of the seeds, spermospheres, shoots, roots and rhizospheres of *Arabidopsis*, *Brachypodium*, maize, wheat, sugarcane, rice, tomato, coffee, common bean, cassava, soybean, switchgrass, sunflower, *Brachiaria*, barley, sorghum and pea. Plants were grown inside sealed jars on sterile sand or farm soil. Seeds and spermospheres contained some uncommon bacteria and many fungi, suggesting at least some of the rare microbiome is vertically transmitted. About 95% and 86% of fungal and bacterial diversity inside plants was uncommon; however, judging by read abundance, uncommon fungal cells are about half of the mycobiome, while uncommon bacterial cells make up less than 11% of the microbiome. Uncommon-seed-transmitted microbiomes consisted mostly of Proteobacteria, Firmicutes, Bacteriodetes, Ascomycetes and Basidiomycetes, which most heavily colonized shoots, to a lesser extent roots, and least of all, rhizospheres. Soil served as a more diverse source of rare microbes than seeds, replacing or excluding the majority of the uncommon-seed-transmitted microbiome. With the rarest microbes, their colonization pattern could either be the result of stringent biotic filtering by most plants, or uneven/stochastic inoculum distribution in seeds or soil. Several strong plant–microbe associations were observed, such as seed transmission to shoots, roots and/or rhizospheres of *Sarocladium zeae* (maize), *Penicillium* (pea and *Phaseolus*), and *Curvularia* (sugarcane), while robust bacterial colonization from cassava field soil occurred with the cyanobacteria *Leptolyngbya* into *Arabidopsis* and *Panicum* roots, and *Streptomyces* into cassava roots. Some abundant microbes such as *Sakaguchia* in rice shoots or *Vermispora* in *Arabidopsis* roots appeared in no other samples, suggesting that they were infrequent, stochastically deposited propagules from either soil or seed (impossible to know based on the available data). Future experiments with culturing and cross-inoculation of these microbes between plants may help us better understand host preferences and their role in plant productivity, perhaps leading to their use in crop microbiome engineering and enhancement of agricultural production.

## 1. Introduction

Modern plants are considered holobionts; an amalgam of different microbes that have coevolved with the host to better survive and cope with numerous biotic and abiotic stresses [1]. Amongst the numerous beneficial plant-associated microbes, the most famous are arbuscular mycorrhizal fungi, which extend through the soil to increase the absorptive area of the root (in 90% of plant species); and nitrogen fixing bacteria colonizing the roots of leguminous plants [2]. Other classical examples of beneficial plant–microbe interactions include the stress resistance conferred to grasses by seed-transmitted *Epichloë* fungi [3] and biocontrol of take-all disease in wheat rhizospheres by antibiotic-producing strains of *Pseudomonas* bacteria [4]. With the extensive technological advances realized in DNA sequencing in the past few decades, an immense diversity of additional plant-associated “difficult-to-culture” microbes have begun to be observed, raising the uncomfortable realization that we do not understand how most of these microbes contribute to the life cycle of the plant, where they came from or the rules of microbial community structuring. Agricultural science’s aspirations to optimize microbiomes, improving crop resilience and productivity, will only be realized if we understand more about the structure, function and provenance of plant microbiomes [5,6,7].

The makeup of microbial populations occupying plants may vary by host genotype, plant age, geographic location, sampling date and tissue type sampled [8,9,10]. The biomes of most interest when studying plants are its rhizosphere, or the soil immediately surrounding the roots; the endosphere inside of the plant, including within and between cells; and the phyllosphere, which includes all the above-ground surfaces of the stem and leaves. Less-studied, seeds have also begun to be understood as important microbial habitats containing diverse bacteria and fungi that can contribute to the microbiome of the next generation of plants [11]. Besides a few examples of vertically transmitted endophytes, traditionally most plant-inhabiting microbes have been believed to derive from soil, passing through the spermosphere and rhizosphere before colonizing the seed, phyllosphere and/or endosphere in much the same way that mycorrhizae and rhizobia do [6,12]. Horizontal transfer of some microbes is also believed to occur as plant surfaces come into contact with insects [13], dust, rain and other plants [14]. Having landed on a plant surface, microbes either have to gain symbiotic access or enter the endosphere through cracks, wounds or stomata [15]. 

With an interest in bioprospecting for agriculturally useful plant-associated microbes, we have attempted to ascertain how and where it is best to search. While soil is clearly an important source of the most well-known beneficial plant-associated microbes [2] we have previously found that maize seeds are also an important source of bacterial endophytes [16], which go on to dominate root endospheres [17] and rhizospheres [18]. Many other publications have likewise been discovering seeds to be rich sources of nonpathogenic bacteria and fungi [11,19,20], which can go on to form significant parts of the microbiomes of tomato, maize, rice, wheat and *Arabidopsis thaliana* [11,21]. It makes intuitive sense that after millions of years of holobiont coevolution [6], plants would ensure that important symbionts are transmitted through their seed to the next generation rather than gambling that these microbes would happen to be present in the soil at the germination site [11]. Some of the important functions these seed-transmitted microbes might provide is aiding in germination, protecting against pathogens, aiding in nutrient acquisition and increasing seedling vigor [11,21,22]. 

Containing multitudes of different microbes, it can be difficult to know where to begin the study of a plant’s microbiome. A common approach to find ecologically important species is to search for a positive correlation between its occupancy and its abundance [23], which has been implemented in microbial ecology to help identify the so called “core microbiome” [24]. Such microbes must have developed a robust transmission strategy, consistently colonizing the plant each generation, contributing to the host species’ growth, survival and/or reproduction [25]. Because of the theoretical importance to agriculture, attempts to find plant core microbiomes have occurred in *Arabidopsis* [26,27], grape [28], potato [29], rice [30], sugarcane [31], switchgrass [32], tomato [33] and wheat [34], to name a few. Theoretically, the common ancestor of angiosperms had a core microbiome before it diverged into monocots and dicots about 150 MYA [35], and this core microbiome may have been transmitted, along with all its attendant ecological functions, to modern plants. Such multispecies core microbiomes have been observed amongst the axenically sprouted seedlings of 28 different crop species [36], amongst roots of various Brassicaceae [37], amongst rhizospheres and roots from 30 species of crop plant grown in soil from surface sterilized seed [38] and in the roots of 31 plant species buried in the sand dunes of a national park in Australia [39]. In a recent publication studying the microbiomes of 17 distinct species of plant raised inside sealed jars, we also found evidence of a core bacterial seed-transmitted microbiome, as well as many bacteria and a few fungi that were common across all plants [20]. 

While core and common members of microbiomes might perform physiological functions that are conserved across plant species, rare or uncommon microbes may also be important to the health and survival of specific plant hosts. This is the outcome of co-evolution between plants, their endophytes and pathogen-specialized processes of signaling, recognition or defense evasion [40], which restrict the microbes to a narrow range of hosts. Legumes, for example, enjoy a symbiosis with specific soil-transmitted rhizobacterial endosymbionts, which use sophisticated molecular signaling to trigger the formation of symbiotic organs in the root where they fix atmospheric nitrogen in exchange for carbon [41]. Beneficial seed-transmitted *Epichloë* fungi are restricted in their symbiosis to certain types of grasses [3], and similarly specific is the symbiosis between orchids and their rhizoctonia-like endosymbionts, without which their seeds will neither germinate nor survive [42]. Grasses (*Dichanthelium lanuginosum*) growing in geothermally heated soils at over 60 °C are able to do so thanks to a fungal endophyte (*Curvularia protuberate*) and its beneficial virus, which if transferred out of their host can also increase the drought and heat tolerance of other plants such as tomato [43]. These host-specific mutualisms/symbiosis are well-known because they produce very strong or ecologically obvious plant phenotypes; however, it is easy to imagine that plant–microbe interactions resulting in more subtle plant phenotypes would be harder to identify. Modern plant microbiome research generates very large amounts of data, which may contain evidence of these subtle and host-plant-restricted plant–microbe associations, but for practical reasons analysis often ends up focusing only on the most common or core microbes. 

This paper aims to document the uncommon (appearing in less than 53% of samples) microbes transferred by seeds or soil and inhabiting rhizospheres, roots and shoots of 17 of the most important angiosperms. These plants include the model plants *Arabidopsis thaliana* (Columbia-0) and *Brachypodium distachyon* (Bd21); the monocot crops rice (*Oryza sativa* ssp. japonica Nipponbare), wheat (*Triticum aestivum*), switchgrass (*Panicum virgatum* Alamo), maize (*Zea mays* ssp. mays B73), sorghum (*Sorghum bicolor* ssp. bicolor), *Brachiaria decumbens*, barley (*Hordeum vulgare* ssp. vulgare) and sugarcane (*Saccharum officinarum*); the dicot crops common bean (*Phaseolus vulgaris* G19833), tomato (*Solanum lycopersicum* Heinz 1706), cassava (*Manihot esculenta*), soybean (*Glycine max*), coffee (*Coffea arabica* Geisha), sunflower (*Helianthus annuus*) and pea (*Pisum sativum*). Plants were grown for up to 2 months inside sealed jars filled with either farm soil or sterile sand, then harvested for DNA extraction from rhizospheres, root endospheres and phyllospheres. Microbiomes of plants raised in soil or sterile sand were bioinformatically contrasted by the sequencing of their PCR amplified fungal ITS and bacterial 16S. Uncommon microbes transmitted by seeds to particular plant species and able to maintain robust populations under agricultural conditions may represent unnoticed but important ecological relationships; information which could help us to fine-tune plant microbiomes for agricultural benefits. 

## 2. Materials and Methods

### 2.1. Sources of Seed

Seventeen different seed accessions were obtained directly from a variety of different providers located on two continents; these are presented in Table 1.

### 2.2. Sources of Substrate

River sand was bought from a hardware store in Palmira, Colombia and manually sifted using a 500-micron metal sieve. 

The soil (a mollisol) used in this experiment was excavated from a cassava field at CIAT near Palmira, Colombia at GPS coordinates 3.498434, −76.354959. Clods were disrupted with crushing, which were then sieved to a uniform consistency using a 500-micron metal sieve.

Autoclaved glass jars that were 7 cm in diameter and 13 cm tall were filled with 100 mL of twice-autoclaved (121 °C for 20 min) sterile sand before a third autoclave treatment, or they were filled with 100 mL of 1:1 soil:sterile sand. 

### 2.3. Experimental Setup and Plant Growth Conditions

Of each plant species, either 0.5 g of small seeds or 20 large seeds were soaked for 6 h in sterile, distilled water within 2 mL or 15 mL tubes. Soaked seeds were then transferred to two sterile Petri dishes containing sterile Whatman #1 filter paper (GE HealthCare: Chicago, IL, USA); seeds in one Petri dish received 3 mL of sterile water, while the other Petri dish received 1 g of field soil resuspended in 3 mL of sterile water. Plates were sealed and incubated at 32 °C for several days in the dark until seeds germinated. 

Once germinated, sterile grown seedlings were transplanted 2 at a time to glass jars filled with sterile sand, while seedlings germinated in soil were transplanted to jars containing a 1:1 blend of soil and sand. After planting, 10 mL of sterile distilled water was poured in and the jars sealed with a plastic lid. Jars were incubated in a Panasonic MLR-352H Plant Growth Chamber set at 28 °C for 12 h with 5 lumens of fluorescent light, and for 12 h at 22 °C of dark. Plants were allowed to grow from 2 weeks to 2 months, until they achieved a significant size or until they hit the lid of the jar. Before plants were harvested, lids were detached inside a laminar flow hood, and plants were permitted to dry off for 24 h. There were 6 unplanted control jars that were watered with 10 mL of sterile water and incubated in the growth cabinet for 14 days: 3 filled with sterile sand, and 3 filled with a mix of sand and field soil. 

### 2.4. Harvesting Rhizospheres, Phyllospheres, Spermospheres, Root and Seed Endospheres

The harvesting of spermospheres and seed endospheres involved either 2 (maize, *Phaseolus*, sunflower), 5 or 0.1 g (*Arabidopsis*, *Brachiaria*, sugarcane) of seeds of each species. These were positioned inside a 15 mL conical tube along with 5 mL sterile, distilled water, then incubated in darkness for 48 h at 32 °C. Tubes were shaken by hand to extricate microbes from the seed surfaces, then supernatant was transferred off into sterile tubes as spermospheres and these were then frozen at −80 °C. The remaining seeds were then surface sterilized/cleaned of DNA by with 30 min of incubation in full-strength bleach (6% Na_2_HPO_4_), then rinsed 3 times in sterile, distilled water and frozen at −80 °C. Two reps of spermosphere and seed endosphere of each species were harvested (68 samples).

For each plant species, 3 repetitions per substrate were sampled by pooling plants inside each jar and then separating them into rhizosphere, root and shoot (306 samples). Using sterile forceps and scissors, phyllospheres were harvested by first removing any remaining seed coat, clipping each shoot just above where it emerged from the substrate, further cutting it into smaller pieces and placing into a clean 50 mL tube, before freezing at −80 °C. Rhizospheres were collected from unwashed roots that had been exhumed, which were then shaken free of loosely attached soil and subsequently relocated into 50 mL conical tubes. To each tube, there was then 10 mL of sterile, distilled water added, followed by vigorous shaking by hand, with the resulting “muddy water” decanted off into a separate 15 mL tubes and labeled as rhizosphere, which was then frozen at −80 °C. After removal of rhizosphere, the roots were washed many more times with sterile, distilled water until both the root surfaces and washwater were clean and clear. Using sterile scissors, clean roots were then cut into pieces within the tube, relocated to a fresh 50 mL conical tube, then frozen at −80 °C.

### 2.5. Sample Preparation and DNA Extraction 

Frozen liquid samples (spermospheres and rhizospheres) were centrifuged at 15,000× *g* for 5 min to concentrate microbial cells as a pellet. Supernatant was taken off and the procedure repeated until 3 mL of sample had been concentrated. The resulting microbial pellet was resuspended in an additional 1 mL of unfrozen rhizosphere or spermosphere. On the other hand, after unfreezing, 1 mL of sterile, distilled water and five 6.35 mm carbon steel ball bearings were added to shoots, roots and seeds in tubes, followed by hand-shaking until the liquid took on the consistency of thick soup. 

A total of 400 uL was taken from these slurries and transferred to 2 mL microcentrifuge tubes containing five 2.3 mm zirconia/silica beads (Cat#11079125z, Biospec Products, Bartlesville, OK, USA) and RNAse A, Phenolics Blocker, and Solution SL 500 uL of Qiagen Powerbead solution (Qiagen, Germantown, MD, USA). For 20 min, these samples were shaken using a Harbil 5G-HD 5 Gallon Shaker (Part#32940, Fluid Management, Wheeling, IL, USA), followed by centrifugation for 2 min at 13,000 RCF. A total of 700 µL of the supernatant was transferred to a fresh tube. The rest of the protocol was followed as per Qiagen instructions with the DNeasy PowerPlant Pro HTP 96 Kit (Qiagen, USA).

### 2.6. Metagenomic Sequencing Library Preparation

16S and ITS amplicons were prepared for sequencing on the Illumina MiSeq platform using a 2-step PCR strategy. First was amplification of all 384 DNA samples using bacterial 16S primers and fungal ITS primers (768 PCR reactions), followed by dual labeling with 6 bp indexes and flow cell adapters. The first PCR was performed using an equimolar mix of staggered, universal fungal ITS (ITS1F and ITS2R [44]) or bacterial 16S (515FB and 806RB [45]) primers containing 19 or 20 bp 5′ tail sequences complementary to Illumina MiSeq indexing primers (Appendix A). Peptide nucleic acid (PNA) blockers against chloroplasts 5′-GGCTCAACCCTGGACAG-3′ and mitochondria 5′-GGCAAGTGTTCTTCGGA-3′ were added to the bacterial 16S PCR reactions to reduce non-target DNA amplification [45]. Reactions were set up in a total volume of 25 uL, including 18.3 µL of nuclease-free water, 4 µL of 5X Phusion^®^ HF buffer, 0.4 µL of each PNA blocker, 0.4 µL of each forward and reverse primer at 10 mM, 0.4 µL of 10 mM dNTPs, 0.2 µL of BSA, 0.1 µL of Phusion^®^ enzyme (NEB, Ipswich, MA, USA) and 0.5 uL of template DNA. The reaction program used was 35X (denaturation at 98 °C for 10 s, PNA annealing at 81 °C for 10 s, primer annealing at 50 °C for 10 s, elongation at 72 °C for 20 s), then a final elongation at 72 °C for 5 min and a cooldown to 4 °C. 

Without evaluating for reaction success, 0.5 uL of PCR product from each of the first 768 PCR reactions was used as template in a second PCR with the purpose of adding dual indexes and flow cell adapter sequences. The 768 distinct labeling reactions were realized using 24 diverse forward primers (TruSeq_F) containing unique 6-basepair-long indexes, and 32 diverse reverse primers (TruSeq_R) containing unique 6-basepair-long indexes (Appendix A). Labeling reactions (step 2) were setup in a total volume of 25 uL, including 19.2 µL of nuclease-free water, 4 µL of 5X Phusion^®^ HF buffer, 0.4 µL of each TSf and TSr primer at 10 mM, 0.4 µL of 10 mM dNTPs, 0.1 µL of Phusion^®^ enzyme (NEB, Ipswich, MA, USA) and 0.5 uL of unpurified PCR product from step 1. Reactions began with 98 °C for 30 s, 15× (denaturization at 98 °C for 10 s, primer annealing and elongation at 72 °C for 20 s), final elongation at 72 °C for 5 min and then a cooldown to 4 °C. 

Amplicons of these 768 labeling reactions were inspected visually for success (bacterial 16S of 428 bp and fungal ITS of 470–525 bp) on 1% agarose gels and estimates of quantity were made with ImageJ [46]. Based on these estimates of amplicon quantity, equimolar amounts of each PCR product within a 96-well plate were mixed into 8 molecular pools (note: except for negative controls which did not amplify, all reactions were repeated until there was sufficient PCR product of each sample). Pooled PCR products were concentrated using ethanol precipitation and resuspended in 10% their volume of pure water. Target amplicons were purified by loading and running 200 uL of each concentrated pool (8 different pools) on a 2% agarose gel, the appropriate bands excised using a scalpel, and gel blocks extracted employing an Omega Bio-Tek E.Z.N.A.^®^ gel extraction kit (Norcross, GA, USA). The 8 purified pools were again checked visually for purity on an agarose gel, quantified using the Picogreen^®^ dsDNA quantitation assay (ThermoFisher Scientific, Waltham, MA, USA) and sent for super-pooling and sequencing on a single 2 × 300 bp paired-end run on the Illumina MiSeq platform at a commercial sequencing facility (GENEWIZ, South Plainfield, NJ, USA). 

### 2.7. Bioinformatics

Data were demultiplexed by the sequencing service provider and received as FastQ files (one per sample). Additional processing involved USEARCH 11 and recommended parameters (www.drive5.com, accessed on 19 June 2019). Briefly, pair-end reads were aligned, then merged, forming full-length sequences known as “uniques”, which were then quality-filtered to remove unmatched and poor-quality reads. Next, the software collected full-length reads together at a similarity threshold of 97%, then created a representative reference sequence for each collection, which is known as an operational taxonomic unit (OTU). OTUs represented by only one raw read were discarded from further analysis. To taxonomically annotate bacterial 16S OTUs, USearch was trained on the RDP training set v16 (13,000 sequences), while fungal ITS OTUs were identified by RDP Classifier [47] trained on the RDP Warcup training set v2 (18,000 sequences). OTU annotations and read counts were exported to Excel (Microsoft, Seattle, WA, USA) for analysis and visualization. Statistics were conducted by XLStat (Addinsoft, Paris, France) and PAST 4 (https://folk.uio.no/ohammer/past/, accessed on 1 July 2019). Based on taxonomic annotation, we manually removed OTUs representing mitrochondria, chloroplasts, plant ribosomes or other nontarget sequences. OTU sequences and annotations are supplied as Appendix A (fungal ITS OTU counts/taxonomy) and Appendix A (bacterial 16S OTU counts/taxonomy). Summed and averaged bacterial 16S and fungal ITS OTU counts are supplied as Appendix A, respectively. OTU counts in seeds and spermospheres are supplied as Appendix A. 

## 3. Results

### 3.1. Sequencing Summary

After removal of chimeras, followed by length, singleton and quality filtering, there were only 3,203,861 high-quality fungal ITS reads to be clustered at 97% sequence identity into 680 OTUs (Appendix A). Manual checking of OTU taxonomy turned up 127 non-target OTUs that had been annotated as plant, protist or bacterial ribosome DNA, so these were also removed, leaving 2,116,837 reads. Of the 377 fungal ITS samples, 375 returned high-quality data, with read counts ranging from 1 for barley sand root #1 to 44,304 for *Panicum* soil-grown rhizosphere #3. Fungal ITS read counts averaged 8058 per sample, although many seed, spermosphere and shoot samples had very low numbers of reads. 

Bacterial 16S read data were also quality-filtered, resulting in 5,370,471 high-quality reads that (at 97% sequence identity) were clustered into 1178 OTUs (Appendix A). Hand-checking of bacterial OTU annotation revealed that despite our use of PNA blockers, 102 non-target OTUs were nevertheless annotated as plant/fungi mitochondria or chloroplast; thus, these were eliminated, leaving 4,945,887 bacterial 16S reads. All bacterial 16S samples contained read count data, ranging from 12 for maize soil shoot #2, to 80,970 for *Brachypodium* spermosphere #2. Bacterial 16S read counts averaged about 13,015 per sample. 

To compare diversity across different shoot, root and rhizosphere samples, we summed reads of each OTU from all repetitions, then transformed the reads into relative proportion for fungi (Appendix A) and for bacteria (Appendix A). Seed and spermosphere bacterial 16S and fungal ITS OTU counts are reported together (Appendix A). The microbial sequence data generated in this study using MiSeq have been deposited and are available in the NCBI Sequence Read Archive (SRA) under BioProject PRJNA731997 and are also provided as Appendix A (annotated sequences and OTU counts). 

### 3.2. Uncommon Microbes in Seeds and Spermospheres

Seeds contained an average of 14 fungal and 56 bacterial OTUs each, while each spermosphere contained an average of 23 fungal and 133 bacterial OTUs. To search for rare/uncommon OTUs amongst this diversity, we screened out all OTUs that occurred in more than half of seeds/spermospheres. A total of 93% of bacterial and 94% of fungal OTUs were uncommon in spermospheres, while in seeds, 95% of both bacterial and fungal OTUs were. Figure 1 shows uncommon and abundant OTUs, providing evidence that seeds can transmit rare microbial propagules, especially fungi. 

### 3.3. Differences in Microbial Diversity between Sample Types

Different plant species, including cassava (Figure 2A) and sorghum (Figure 2B), were grown in jars containing sterile sand or field soil. To relate the microbial diversity in different tissue types, OTU counts (summed and averaged across reps) were transformed into presence–absence data, then ordinated by principle component analysis of covariance (n − 1) and visualized as scatterplots grouped by tissue type and substrate (Figure 2C,D). The number of different OTUs per sample was also plotted by substrate and tissue type, allowing for visualization of differences in diversity amongst bacterial (Figure 2E) and fungal (Figure 2F) populations. The most obvious separation between groups is evident for both bacteria and fungi in rhizospheres of soil-grown plants, which are clearly distinct from sterile sand ground plants. About half of either bacterial or fungal root samples from plants grown on soil were also observed to shift away from sterile sand root samples. There was no clear difference in PCAs for either bacterial or fungal diversity in sand and soil grown shoots. 

The strong shift in rhizosphere microbial diversity is easy to see when plotting numbers of bacterial (Figure 2E) and fungal (Figure 2F) OTUs next to each other, where mean values in sand went up from 252 and 25, respectively, to 594 and 127 when grown on soil. There were only modest average increases in the number of OTUs that are gained by bacterial or fungal populations in roots when grown on soil relative to sterile sand, going from 177 and 24 on sand to 225 and 37 on soil. While the average number of fungal OTUs in shoots increased slightly when plants were grown on soil instead of sand (from 16 to 24), the average number of bacterial OTUs in shoots instead went down when plants were grown on soil rather than sand (from 255 to 185). 

To obtain a taxonomic summary of all uncommon OTUs, all reps of each treatment (e.g., bacteria in soil-grown maize roots) were summed together, normalized to percentage, screened out if present in more than 9/17 plant species, then averaged across tissue and soil type (e.g., all uncommon fungi in sand-grown shoots). Figure 3 shows the total number of uncommon fungal OTUs (Figure 3A) or uncommon bacterial OTUs (Figure 3C), as well as the average number of uncommon reads for fungi (Figure 3B) and bacteria (Figure 3D). First of all, it is important to note that all plants, whether grown on field soil or sterile sand inside hermetically sealed jars, nevertheless developed microbiomes, showing that seeds can transmit diverse microbes to the plant. Sterile-grown shoots contained the lowest number of fungal OTUs with only 92 (96% of total), which increased to 140 (96% of total) in roots and 155 (95% of total) in rhizospheres. Judging by read abundance, these uncommon, apparently seed-transmitted microbes made up a very large proportion of the fungal population, totaling 51, 52 and 52% of the reads observed in sterile sand-grown shoots, roots and rhizospheres (Figure 3B). When plants were grown on soil, the total number of uncommon fungal OTUs observed in shoots, roots and rhizospheres went up to 126 (95% of total), 203 (95% of total) and 399 (87% of total) respectively (Figure 3A). Although the total number of uncommon fungal OTUs went up when grown on soil, their average abundance went down, making up only 30, 42 and 26% of the total reads in shoots, roots and rhizospheres (Figure 3B). The majority of fungal OTUs and reads observed in all plants and sample types were Ascomycetes followed by Basidiomycetes. It was interesting to note the presence of Glomeromycota (mycorrhizae) and Chrytidiomycota (root pathogens) in soil-grown roots and/or rhizospheres, although they were not represented by a large number of reads. 

There was a substantial number of uncommon bacteria (478 OTUs representing 72% of the total) in sterile-grown shoots that must have been transferred there through seeds, and growth in soil did not dramatically increase this number (525 OTUs representing 84% of the total). Similarly, a total of 499 (82% of total) uncommon bacterial OTUs were observed in sterile sand-grown roots, which increased to 647 (88% of total) in soil-grown plants (Figure 3C). Surprisingly, growth on soil reduced the number of uncommon bacterial OTUs from 486 (73% of total) in sterile sand-grown plants to 351 (38% of total). Because numbers of uncommon bacterial OTUs were usually about 3 times higher than those of uncommon fungal OTUs, it was surprising to see that these represented very little (less than 10%) of the average read abundance in a sample (Figure 3D). Uncommon bacterial reads in sterile sand-grown shoots represented only 2% of the total, while in roots they were 6% and in rhizospheres 3%. Growth on soil vs. sterile sand approximately doubled the proportion of uncommon bacterial reads in shoots, going on average from about 2% to 5% and from about 6% to 11% in roots, while in rhizospheres, soil caused a drop to only 1%. The most abundant uncommon bacteria were Proteobacteria and Firmicutes, but in soil-grown roots it was Cyanobacteria that dominated, representing 5% of the total reads. 

In order to observe the stability of seed microbiome transmission to plants, we calculated the Sørenson similarity index by comparing the occurrence of every uncommon OTU between field soil- and sterile sand-grown plants (Table 2). FungalOTU32, for example, was found in maize roots grown on both sterile sand and soil; thus, it was considered seed-transmitted and raised the corresponding index. By this metric, seed transmission appears to have been responsible for only a small amount of the uncommon diversity observed in soil-grown plants, contributing on average only 8 and 15% of the fungal and bacterial diversity in rhizospheres, 22 and 24% of the fungal and bacterial diversity inside roots, and 21 and 26% of the fungal and bacterial diversity in shoots. Because soil-grown plants had both relatively few seed-transmitted OTUs and a higher number of uncommon microbes than did sterile sand-grown plants, this potentially means that propagules of uncommon-seed-inhabiting microbes are stochastically distributed amongst seeds and/or that the majority of uncommon-seed-transmitted microbes are quickly displaced by soil-transmitted ones (i.e. most uncommon plant microbiome inhabitants are soil-transmitted). It is also worth noting that there is substantial variation in plant microbiomes depending on what substrate they were grown on, with, for example, 0% of uncommon fungi in *Brachypodium* shoots being observed on both substrates, while there was a similarity of 51% between uncommon fungi in soy roots. Other samples with relatively high similarity between substrates were fungi in maize/rice roots and maize/soy/tomato shoots, bacteria in Arabidopsis/sugarcane/tomato roots and Arabidopsis/Brachypodium/pea/tomato shoots. There was much more similarity between roots and shoots (0.21–0.26) grown on different substrates than there was for rhizospheres (0.08–0.15). 

### 3.4. Patterns in Microbial Abundance Data

In an effort to visualize patterns in the abundance and distribution of uncommon microbes amongst tissues/plant species/substrates, reads were added together across samples, normalized to proportional abundance, and the top 40 most abundant but uncommon OTUs were displayed as heat maps sorted by Bray–Curtis dissimilarity indicated by hierarchical trees (Figure 4). Groupings by plant are indicated with a purple bar across the top. 

Focusing on bacterial squares shaded dark red (greater than 5% read abundance) there are none in shoots, 3 in roots and only 1 in coffee rhizospheres. Three other interesting patterns are evident depending on clustering of vertical dark blue lines (indicating no reads in any OTU). Nine soil-grown samples, including *Phaseolus*, soy, rice, maize and wheat, had a near absence of uncommon OTUs in shoots, suggesting that soil bacteria somehow reduced total bacterial diversity in above-ground tissues, apparently by enhancing the abundance of common bacteria instead. Conversely (and more expectedly) roots of eight different sand-grown plants had lowered diversity of uncommon bacterial OTUs suggesting that a lack of inoculum resulted in a reduction in bacterial endophytes. Most expectedly of all, rhizosphere populations of uncommon bacteria appear to be strongly influenced by inoculation with soil, with all 17 samples clustering together by substrate. Clustering by plant rather than by soil might indicate seed transmission of microbes, and the greatest number of instances of this phenomenon were observed in shoots where *Bracharia*, coffee and pea grouped together, while in roots there was only two groupings (*Brachypodium* and soy) and in rhizospheres only one (maize). 

Fungal heatmaps also show soil’s importance in populating rhizospheres with uncommon microbes (e.g., FungOTU32, 52 and 88), with most samples (except sorghum, maize and sugarcane which grouped by plant species) clustering by substrate type. Neither root, nor shoot populations of fungi appeared to be strongly influenced by substrate microbiology; however, for roots there were eight groupings by plant (almost half), suggesting that vertical transmission of uncommon fungi to roots is significant in many plant species. Another interesting pattern in fungal heatmaps was single dark-red fungal OTU squares occuring only once per row; for example, FungOTU58 appeared in sand-grown *Bracharia* shoots but not at all in soil-grown shoots, while FungOTU76 occurs at >5% in soil-grown *Arabidopsis* roots but not at all in any other root sample, suggesting that either seeds or soil (or both) are serving as a source of stochastic inoculum. Another interesting pattern was of two red or colored squares side by side, showing seed transmission of fungus to the shoots or roots under both sets of growing conditions. Examples include FungOTU32 in coffee/maize roots; FungOTU16 in *Arabidopsis*/coffee roots; FungOTU166 in soy roots, FungOTU338 in sugarcane shoots; and FungOTU170 in coffee shoots

### 3.5. Taxonomy of Uncommon yet Abundant Microbes

In order to better define which uncommon fungi (Figure 5) and bacteria (Figure 6) might help to distinguish one plant microbiome from another, we again selected OTUs present in 9 or fewer of the 17 sterile sand- or soil-grown samples, and show the most abundant in each sample type. 

The majority of uncommon shoot mycobiomes were represented by a few dominant OTUs, whether plants were grown on sterile sand or field soil. For example, 75.9% of the fungal reads in sand grown cassava shoots were contributed by FungOTU101 (*Hanaella*) and 104 (*Bullera*), which occurred in no other plant under any condition (suggesting stochastic seed-borne inoculum); however, when grown on soil, the uncommon cassava shoot mycobiome was 43% FungOTU22, 27, 28 and 55 instead. Sterile sand-grown tomato shoots were dominated (89.4%) by the fungus *Punctulariopsis* (FungOTU45), 99.8% of reads in *Phaseolus* belonged to FungOTU5 (*Penicillium*), and 77.5% of fungal reads in *Brachiaria* were from FungOTU24 (*Papiliotrema*)—when grown on soil, only *Phaseolus* shoots maintained an uncommon fungus at a level of above 5% (78% for FungOTU19 and 10% for FungOTU91). In sterile sand, the most abundant, uncommon fungus in rice was FungOTU14 (*Alternaria*), but on soil it was replaced by FungOTU80 (*Sakaguchia*) at 95%. Although they were not technically uncommon, it is worth noting that on soil, the dominant members making up more than 99% of reads in maize and pea shoot mycobiomes were seed-transmitted FungOTU9 (*Sarocladium*) and FungOTU5 (*Penicillium*), respectively. The magnitude of OTU abundance in roots was generally lower than that of shoots, with only six examples near or above 50% in sterile sand and one in soil, and these included 49% of reads in cassava roots belonging to FungOTU35 (*Aspergillus*); 75% of reads in *Brachypodium* coming from FungOTU37 (*Chaetomium*); 54% of reads in maize coming from FungOTU41 (*Aspergillus*); 62% of reads in *Panicum* roots coming from FungOTU62 (*Phoma*); 48% of reads in sugarcane roots growing in sterile sand coming from FungOTU68 (*Ustilago*); and between FungOTU36 (*Myrothecium*) and 42 (*Epicoccum*), 93.9% of reads in *Brachiaria* roots were accounted for. Although most soil-grown roots contained one or two abundant and uncommon fungal OTUs, only FungOTU19 (*Penicillium*) in *Panicum* was recorded at an abundance above 50%, and it appeared to have been stochastically transmitted through seed. OTUs such as #28 (*Mortierella*) and #76 (*Plectosphaerella*) appear to have been soil-transmitted since they were detected in all soil-grown rhizospheres; however, they appeared inside few roots, suggesting that most plants were able to block them from entering. Sterile sand-grown rhizospheres were dominated by a few hyperabundant fungi; examples included 99% of the reads in coffee coming from FungOTU10 (*Nectria*), 73% of the reads in pea coming from FungOTU12 (*Alternaria*), 54% of the reads in *Bracharia* coming from FungOTU36 (*Myrothecium*), 93% of the reads in *Brachypodium* coming from FungOTU37 (*Chaetomium*) and 62% of the reads in *Panicum* were from FungOTU 62 (*Phoma*). Almost all soil-grown rhizospheres contained dominant uncommon fungi, but only the seed-transmitted/biotically filtered FungOTU9 (*Sarocladium*) in maize occurred at a level of over 50%. 

Contrary to fungi, there were almost no both uncommon and abundant bacterial OTUs in any of these samples; thus, we raised the threshold for screening out an OTU from 0% to 0.15%, allowing for the inclusion of additional and potentially interesting OTUs. Even with this elevated detection threshold, no bacterial OTU in sterile-grown shoots was observed representing more than 5% of the total reads. When grown on soil however, *Brachiaria* shoots did contain abundant BactOTU21 (*Xanthomonas*); coffee shoots contained BactOTU7 (*Massilia*); sorghum shoots contained abundant BactOTUs 29, 32 and 101; and both sunflower and tomato shoots contained abundant BactOTU34 (*Delftia*). Eight plant species had roots with abundant bacteria in sterile sand, and eight did in soil; in sterile sand, these included coffee with *Rhizobium* (20% of BactOTU30) and *Methylobacterium* (19% BactOTU28) and sugarcane with *Luteibacter* (20% of BactOTU79) and *Rhizobium* (8% BactOTU758). While soy or pea roots grown in soil had been expected to accumulate large amounts of rhizobial bacteria, they instead accumulated abundant and uncommon OTUs of *Pseudomonas*. Interestingly, *Arabidopsis* and *Panicum* roots grown on soil accumulated significant amounts of *Leptolyngbya* bacteria, which they did not in sterile sand. Cassava roots grown in the cassava farm soil also accumulated *Streptomyces* and *Pseudoduganella* that were not abundant in any other plant. Nine sand-grown rhizospheres contained uncommon bacteria present at a level above 5% of the total reads, with the highest being BactOTU17 (*Pseudoxanthomonas*) at 33% in sorghum and BactOTU30 (*Rhizobium)* at 22% in coffee. In soil-grown rhizospheres, only BactOTU33 (*Pseudoxanthomonas*) and BactOTU46 (*Chitinophaga*) in barley; BactOTU62 (*Desomontoc*) in coffee appeared at relative abundances above 5%. 

## 4. Discussion

Nearly all angiosperms use seeds for their reproduction, and as such, it would make sense that they also serve as vectors for ecologically and evolutionarily important members of their microbiomes. We previously showed that seeds transmit a dominant and common/core microbiome to angiosperms [20], but we did not observe whether uncommon/rare plant microbiomes are similarly transmitted. The rare microbiome usually refers to microbes present at low levels of abundance in a population, but in the context of our study involving many different plant species, we use the words “rare” and “uncommon” interchangeably to refer to microbes that have low rates of occupancy across plant hosts, irrespective of their abundance. Because uncommon microbes are often overlooked in microbiome evidence for species-specific symbiosis or horizontally transmitted microbial mutualisms may likewise be overlooked. For example, we would expect to see only soil-grown legumes such as pea and soy accumulating significant levels of rhizobial bacteria in their roots, meaning that they would likely be classified as uncommon and ignored during a search for core plant microbiomes. While studying uncommon microbes, we decided to focus on relatively abundant OTUs because of the macroecological mass–ratio hypothesis, which predicts that the effect of a species on ecosystem function is proportional to its relative abundance [48,49], while understanding that scarce microbes could be important as well [50]. We hope to begin answering such questions as: Are there uncommon seed- or soil-transmitted microbes that accumulate to high levels inside particular plant species, suggesting that they contribute to the varied ecological adaptations of different plants? Do these uncommon seed-transmitted microbes get displaced by soil-transmitted microbes, and if so, by which ones? Are certain microbes infrequent members of the plant microbiome because of low or stochastic rates of transmission? Using high-throughput sequencing, we identified and quantified bacteria and fungi found in these samples, focusing only on abundant (greater than 5% of the total reads in a sample) and “uncommon” OTUs, which were found in 9 or fewer of the 17 plant species. Seeds and spermospheres were found to contain a few bacteria and many fungi that fit this criteria, suggesting that a significant amount of the variability in plant microbiomes might begin with the seed. 

### 4.1. Uncommon but Abundant Microbes in Rhizospheres

The layer of soil immediately around the root is the rhizosphere, where large populations of up to 10^11^ microbial cells/gram conduct ecological functions beneficial for controlling disease and aiding in nutrient absorption of the plant [51]. Roots manipulate the rhizosphere microbiome by attracting and feeding microbes via a plethora of released sugars, phytosiderophores, organic acids, amino acids, vitamins, mucilage and nucleosides [14]. Because plants invest so much energy in attracting and feeding soil microbes, it has traditionally been assumed that the entire rhizosphere microbiome “is recruited from the main reservoir of microorganisms present in soil” [12]. Publications about rhizosphere microbiomes echo this idea, suggesting that most or all rhizosphere microbes in *Arabidopsis* [26], maize [52], coffee [53], rice [30], common bean [54] and soy [55] are soil-derived. Seeds are increasingly being shown to contribute microbes to the rhizosphere as well. Endophytic bacteria from seeds can travel systematically through the plant, exit roots and colonize the rhizosphere [16], perhaps depending on root hair expulsion [56] or by being sloughed off inside root cap border cells [57]. We have recently shown that the most abundant bacteria and fungi in juvenile plant rhizospheres are in fact seed-transmitted, while soil does transmit a large diversity of common microbes to this niche as well [20]. In the current study, uncommon fungal OTUs, which were transmitted by seeds to the rhizosphere of sterile sand-grown plants made up 50% of the mycobiome by abundance (Figure 3), but on average only 5% of these uncommon fungal OTUs went on to establish themselves in soil-grown rhizospheres (Table 2). While soil-grown plants had the higher number of uncommon microbes in their rhizospheres, the average total abundance of uncommon fungi went down to only 27%; this might be because soil also added many abundant, common fungi to rhizospheres. There were hundreds of seed-transmitted, uncommon bacteria that were observed in sand-grown rhizospheres, making up a very small part of the population by abundance, but growth on soil only allowed about 10% of these to colonize the rhizosphere while further reducing uncommon bacteria in number and abundance; again, this might be because soil also added many abundant, common bacteria to rhizospheres. 

With the exception of maize, sorghum and sunflower rhizospheres which were mostly common fungi, all plants grown in sterile sand developed rhizospheres dominated by rare fungi. In many cases, these uncommon fungal OTUs represented nearly the entire rhizosphere mycobiome, for example, 93% of the reads in *Brachypodium* coming from FungOTU37 (*Chaetomium*) and 99% of the reads in coffee coming from FungOTU10 (*Nectria*). Because rhizospheres of sterile sand-grown plants were not very diverse, containing on average only 25 OTUs, these fungi could represent stochastically selected “island” colonists or founders that rose to dominance through lack of competition [58]. Indeed, FungOTU10 and FungOTU37 were absent from soil-grown rhizospheres of these plants, suggesting either that they were not present on the particular seeds planted in soil, or that these fungi were outcompeted by more aggressive soil rhizosphere colonists. A microbe abundant in both sterile sand and soil-grown rhizospheres might be an example of a specially adapted, seed-transmitted symbiont, and there were three of these: FungOTU70 (*Bipolaris*) appeared in high abundance (6–39%) in both sterile sand- and soil-grown rhizospheres of *Brachiaria*; while in sugarcane, FungOTU15 and FungOTU338 (both *Curvularia*) appeared at high abundance in both sterile sand- and soil-grown rhizospheres. These fungi (*Bipolaris* and *Curvularia)* as well as *Chaetomium, Penicillium, Phoma, Alternaria, Aspergillus, Cladosporium, Colletotrichum, Fusarium* and *Rhizopus* are considered seed-transmitted pathogens of *Brachiaria* and *Panicum* [59]. *Curvularia* is also known as a beneficial endophyte of tropical panic grasses, where its presence protects the host against extreme temperatures [43]. While not technically uncommon by our definition, FungOTU9 (*Sarocladium zeae*) in maize was also found in high abundance in both sterile sand- (12%) and soil-grown rhizospheres (88%), suggesting a special seed-transmitted symbiosis. *Sarocladium zeae* (also known as *Acremonium zeae*) is a well-known seed-transmitted endophyte of maize that produces a range of insecticidal and antibacterial compounds which it deploys to help protect its plant host [60]. As an endophyte with biocontrol potential, perhaps *Sarocladium zeae* is also able to protect maize against herbivory and infections in the rhizosphere. 

Some abundant/uncommon fungi were only observed in soil-grown rhizospheres, for example, FungOTU38 (*Penicillium*) in pea, or FungOTU51 (*Hyphodermella*) in *Phaseolus*/*Panicum*. *Hyphodermella* is a white rot fungus that associates with decaying plant material [61] and has been isolated as an endophyte in wheat [62], but it is difficult to understand why it accumulated to high levels in some rhizospheres and not others. *Penicillium* has often been reported in rhizospheres where it can solubilize phosphate and increase its uptake by pea [63] or promote millet defense and growth [64]. We had assumed that soil would contain high densities of evenly distributed fungal inoculum, however it is possible that many fungal propagules are rather scarce in soil, resulting in random distribution from soil sample to sample. For context, arbuscular mycorrhizal spores in Indian sugarcane fields have been recorded at a density of as little as 1.19 per gram of soil [65], Fusarium wilt of spinach has been observed to require at least 10 propagules per gram of soil [66] and microsclerotia of *Verticillium* have been found to vary from 0 to over 400 per gram of soil within a single field [67]. It is thus easy to imagine that if fungal inocula were present at lower densities, say 0.5 spores per gram, our filling of different jars with 50 mL of soil could have resulted in stochastic fungal distribution and unequal inoculation of rhizospheres. OTUs such as FungOTU51, 78 and 92 were only present in one or two soil-grown rhizospheres and are examples of soil-transmitted fungi with low inoculum levels that were unevenly distributed into different soil-filled jars. Conversely, *Penicillium, Chaetomium* and *Curvularia* (as well as *Alternaria, Aureobasidium, Cladosporium, Fusarium* and *Acremonium*) have been shown to be cosmopolitan to soils around the world [68], so these genera might have been expected to occur in all soil rhizospheres (as we previously observed for *Fusarium* and *Alternaria* [20]) rather than only one or two. Fungi that were observed in all soil, but not sterile sand-grown rhizospheres, would arguably be present at high enough propagule densities to serve as a uniform inoculum for plants growing in all jars. The best examples of this were fungal OTUs 52 (*Alternaria*), 66 (*Tomentella*), 85 (*Xylogone*), 86 (*Mortierella*) and 88 (*Saitozyma*), which appeared in either 16 or 17 of the soil-grown rhizospheres but nearly zero sterile sand-grown rhizospheres. 

Because there were almost no “uncommon and abundant” bacteria in any sample type, it was necessary to lift the minimum threshold of detection to ignore any OTU with less than 0.15% of the reads in a sample. With this relaxed threshold, nine sterile sand-grown rhizospheres had abundant, uncommon bacteria, while only two plants grown in soil did. In these two soil-grown samples, there were three uncommon/abundant OTUs that were absent from sand-grown rhizospheres, suggesting that they could have been specially selected and enriched by the plant. Interestingly, the average abundance of uncommon bacteria in rhizospheres went down when grown on soil, which might be explained if soil supplies more high density, common bacteria to the rhizosphere rather than scarce and rare propagules, reducing the stochastic effects of segregating soil into different jars. In general, however, under the conditions of our experiment, uncommon bacteria appear to be numerically insignificant in rhizospheres, being reduced in both diversity and abundance when challenged with high densities of common soil bacteria. While this appears to suggest that there is little or no specialized coevolution between most of these crop plants and rhizobacteria found in this cassava field soil, it also reinforces our previous observation that plant rhizospheres are dominated by common seed-transmitted bacteria [20]. 

### 4.2. Uncommon but Abundant Microbes in Roots

Roots anchor the plant in the substrate, where they also absorb water and nutrients while secreting biochemicals to attract, communicate and cooperate with soil microbes; some of which have typically been assumed to invade and populate the root as endophytes [6]. Root microbiome variation between plant genotypes is well-established [37,38,39,69,70,71,72] and is usually explained as variation in the plant’s biological filtration abilities, which restrict or promote soil microbe entry. Assuming a soil origin for most of the root microbiome, many studies that sterilize their seeds and leave out sterile substrates as negative controls nevertheless continue to conclude that endophytes come from the soil. For example, growing all plants in microbe-filled soil without a sterile substrate for comparison, the recruitment of *Brassica napus* seedling microbiota was deemed to derive almost entirely from soil or other unknown sources rather than seeds [73]. To the contrary, we have shown that seeds of all plants tested transmit microbes to offspring, and most of these microbes go on to dominate the microbiome of juvenile plants [20]. That being said, these seed-transmitted microbiomes may fluctuate in makeup over time, increasing in diversity during germination [36] and later decreasing as plants age [74]. Ordination of microbiome data in this study suggests that about half of plant species receive the majority of their root bacterial and fungal diversity from seeds rather than soil (Figure 2C,D), while soil also increased the total number of uncommon OTUs in all plants (Figure 3A,C). Uncommon fungi represented nearly all of the diversity inside roots (95%); however, growth on soil slightly reduced their average relative abundance to 42%, which suggests that some common soil-transmitted fungi displaced uncommon seed-transmitted fungi (Figure 3A). While uncommon bacteria represented up to 88% of the diversity in roots, they were of scarce abundance, going from 7% in sand up to a maximum of only 11% when plants when grown in soil (Figure 3C). There were very few both uncommon and abundant bacterial OTUs in roots, and they were transmitted by seeds (Figure 4). Conversely, there were many instances of abundant and uncommon fungal OTUs found in nearly every root whether grown in sterile sand or soil, suggesting both seed and soil transmission of uncommon fungi. Distribution of fungal propagules seemed to vary stochastically, making it difficult to know whether detection inside a root was a product of biotic filtering by the plant or chance inoculation. 

Most bacterial root endophytes are Actinobacteria, Bacteroidetes and Proteobacteria [14] of the genera *Acidovorax, Agrobacterium, Arthrobacter, Bacillus, Curtobacterium, Enterobacter, Erwinia, Methylobacterium*, *Micrococcus*, *Phyllobacterium, Pantoea, Pseudomonas, Rhizobium, Serratia, Stenotrophomonas*, *Streptomyces* and *Xanthomonas* [75]. We observed uncommon and abundant BactOTUs 39 (*Luteibacter*), 61 (*Cohnella*), 109 (*Phormidium*), and 758 (*Rhizobium*) in roots of both sand- and soil-grown plants, suggesting seed transmission. After raising the threshold of detection to include more candidate OTUs for analysis, there were eight sterile sand-grown roots (*Arabidopsis*, *Brachiaria*, *Brachypodium*, coffee, rice, sorghum, switchgrass and sugarcane) where we also observed uncommon/abundant strains of *Methylobacterium*, *Fictibacillus, Cohnella, Herbaspirillum, Luteibacter, Pseudomonas, Paenibacillus, Ralstonia, Mucilaginibacter* and/or *Rhizobium* (in coffee and sugarcane). Some seed-transmitted bacteria increased in abundance in soil-grown roots, including BactOTU63 in coffee, BactOTU39 in *Phaseolus*, BactOTU78 and BactOTU410 in rice and BactOTU185 in sunflower. Of these apparently soil-derived bacteria, switchgrass and especially *Arabidopsis* (52% of its root microbiome) accumulated significant amounts of *Leptolyngbya*, which they did not in sterile sand. *Leptolyngbya* are filamentous cyanobacteria colonizing rhizospheres, where they produce a variety of metabolites such as auxins, which are beneficial in pea, rice [76] and wheat [77] agriculture. Cassava roots in field soil accumulated 20% of their bacteria reads as BactOTU57 of the genus *Streptomyces*, bacteria that are known to be beneficial to tomato and maize growth and stress resistance [78,79]. Because the soil we used came from a field used continuously to grow cassava for many years, it is possible that this abundant *Streptomyces* represents a cassava-specific strain that has been built up over the years in much the same way that protective strains of *Pseudomonas* build up year after year of planting wheat, until they can suppress take-all disease [4]. While soil grown soy or pea roots had been expected to accumulate large amounts of rhizobial bacteria, they surprisingly did not, perhaps because there were no compatible strains in the field. This was the situation for soybeans when they began to arrive in the Americas: there were not any compatible symbionts in the naive soils when soybeans began to arrive in the early 1800s, necessitating first the practice of infected soil transplants from farm to farm and later their specialized inoculation with pure bacterial strains [80]. 

In nature, root endospheres are generally dominated by *Ascomycetes* (*Pezizomycetes*, *Dothideomycetes*, *Sordariomycetes*, *Eurotiomycetes* and *Leotiomycetes*), *Basidiomycota* (*Polyporales*, *Russulales* and *Agaricales*) and Zygomycota [81], although in grasslands, monocot roots are dominated by *Dothideomycetes* (i.e. *Fusarium* and *Alternaria*), while in forests it is Leotiomycetes that predominate [82,83]. The soil-dwelling Chrytidiomycete *Olpidium* can also make up a large portion of root mycobiomes in tomato [84], *Arabidopsis* [85], melon [86] and lettuce [87]. In our experiment, uncommon and very abundant genera in roots of seed-transmitted fungi included *Aspergillus, Chaetomium*, *Embellisia, Epicoccum, Myrothecium* and *Ustilago*; however, when grown on soil, the most abundant fungi changed to *Curvularia, Penicillium, Plectosphaerella* and *Preussia*. The most famous seed-transmitted endophytes are *Epichloë, Acremonium (Sarocladium)* or *Neotyphodium* in grasses [3], but there was evidence for other uncommon/abundant seed-transmitted root fungi in many plants here as well, such as FungOTU16 (*Waitea*) in *Arabidopsis*, FungOTU32 (*Talaromyces*) in maize, FungOTU166 (*Sidera*) in soy, FungOTU15 and 338 (*Curvularia*) in sugarcane together representing 87% of reads, FungOTU37 (*Chaetomium*) in *Brachypodium* and FungOTU19 (*Penicillium*) in *Phaseolus*. Although generally assumed to derive from the soil (we observed it to come from seed), *Waitea* can occur at high levels inside the roots of *Arabidopsis* flavonoid mutants [88] and *Chaetomium* has been observed to be the most abundant fungus in Australian-grown rhizospheres of *Brachypodium* [89]. *Talaromyces* has previously been isolated from maize seed [90]. The occurrence of other uncommon fungi was harder to explain, perhaps because inoculum on seeds was uneven, making it difficult to know if root colonization was specifically allowed by that plant or whether it was the result of a chance inoculation by a wayward propagule. The most extreme example of a rare, stochastically distributed propagule was FungOTU283, a nematophagous fungus of the genus *Vermispora* that was abundant in sterile sand-grown *Arabidopsis* roots but was observed nowhere else in any sample type. 

Seeing abundant microbes in soil, but not sand-grown roots, was evidence of soil transmission. Most fungal endophytes of roots are thought to come from soil, and likewise to be very sensitive to soil biogeography [71,83,85], but we eliminated this variability by sterilizing the sand and using field soil from only one location. Despite thoroughly homogenizing soil, we were apparently unable to achieve an even distribution of rare fungal propagules into each jar; a fact that became evident when comparing rhizospheres. For example, FungOTU90, which belongs to an obligately lichenicolous microbe of the genus *Abrothallus* [91], was only observed in coffee roots and rhizospheres, apparently being so sparsely distributed in the soil that only one of the jars where coffee was planted received inoculum. Conversely, FungOTU23 (*Preussia*) appeared in 4 different soil-grown roots, while being detected in 12 soil-grown rhizospheres—an example of a common soil-transmitted fungus being filtered out by roots. Another example of a fungus that made it past plant biotic filters to colonize roots, FungOTU76 (*Plectosphaerella*) appeared in 12 of the 17 soil-grown rhizospheres, but was only detected inside *Arabidopsis* roots where it made up 40% of the mycobiome. *Plectosphaerella cucumerina* is a soil fungus that can enter soybean stems as an endophyte [92] and that can infect *Arabidopsis* leaves as a necrotroph [93]. 

### 4.3. Uncommon but Abundant Microbes in Shoots

Microbes inside the aerial parts of the plant can influence source/sink relationships, the flux of sugars and nutrients and can even fix nitrogen [94], while in the phylloplane, they could affect the harvesting of light and gas exchange or help prevent the establishment of pathogens. Despite having an incomplete sense of where shoot-inhabiting microbes come from [14], they are usually believed to be horizontally acquired from rain, dust, soil, and contact with other organisms, followed by biotic filtering that shapes the resulting microbiomes over time [95,96,97,98,99]. Speaking to the high diversity of endophytic fungi in coffee shoots, many of these environmentally transmitted microbes have been described as “accidental tourists” playing no direct role in the life cycle of the plant [100]. Some evidence suggests that transmission from seeds to shoots can be significant: maize seeds transmit fungi to the leaves [101], rice seeds populate their shoots with bacteria [102], oak embryos within the seed are heavily colonized by microbes [103], and we have previously shown that juvenile shoot microbiomes are dominated by common/core seed-transmitted bacteria and fungi [20]. Analyzing the total microbiome data in this manuscript again, we found that soil did not greatly affect the diversity of either bacteria or fungi in any shoot sample, implying that most microbes in shoots came from seeds (Figure 2). Focusing exclusively on uncommon microbe abundance in soil-grown plants (Figure 3), bacteria made up only a small amount (5%) of the shoot microbiome, whereas uncommon fungi were about 30% of the shoot mycobiome. More so than in rhizospheres or roots, seeds appear to supply a substantial number of microbes to (soil-grown) shoots, representing 35% of the bacterial OTUs and 19% of fungi (Table 2). Shoot microbiomes had a lower number of uncommon microbes than roots or rhizospheres, suggesting that perhaps plants were able to impose tighter controls on microbial invasion of their distal tissues and organs. The ability of shoots to effectively filter out undesirables is exemplified by maize and *Bromus tectorum*, which are both able to maintain an unvarying leaf mycobiome despite being grown on different soils that are known to contain distinct endophytic fungal communities [72,101]. In addition to filtering, shoots may be able to become saturated by core bacterial endophytes [20] which could serve to outcompete and block less-common microbes from invading the niche.

Plant shoots are usually reported to contain Proteobacteria, Firmicutes, *Actinobacteria* and *Bacteriodetes* with such genera as *Pantoea, Pseudomonas, Bacillus, Sphingomonas, Erwinia, Acinetobacter, Gluconobacter* and *Xanthomonas* [104]. We observed most of these bacteria, plus others such as *Klebsiella* and *Massilia*, as part of the common or core population of shoots of our experiment [20], but strictly speaking, there were no bacteria that were both uncommon and abundant in any shoots. Even with our relaxed detection threshold for bacteria (Figure 6), there were still no uncommon and abundant bacteria in sand-grown shoots, and only seven of the genera *Xanthomonas, Massilia, Delftia, Sphingomonas, Chryseobacterium* and *Ramlibacter* were abundant inside five different soil-grown shoots. In sorghum, the three uncommon and abundant OTUs total 49% of the total reads, while BactOTU21 is 24% of the shoot microbiome of soil-grown *Brachiaria*; however, it otherwise seemed that shoot microbiomes were mostly made up of common bacteria. 

Fungal genera reported in the literature to occur in shoots include *Alternaria, Acremonium, Penicillium, Cladosporium, Mucor, Sporobolomyces, Rhodotorula, Cryptococcus* and *Aspergillus* [104]. We previously observed *Fusarium, Alternaria, Pseudozyma, Sarocladium, Phoma* and *Penicillium* as common fungi in angiosperm shoots [20], but the list of uncommon and abundant shoot fungi that we observed is much larger, also including the genera *Waitea, Sakaguchia, Punctulariopsis, Curvularia, Papiliotrema, Phoma, Hannaella, Nectria, Bullera, Cryptococcus, Sidera, Embellisia, Rhizopus, Ustilago, Chaetomium, Nigrospora, Setophoma, Mortierella, Davidiella, Occultifur, Clonostachys* and *Aspergillus*. Perhaps because fungal diversity in shoots was so low, with an average of only 16 OTUs in sand-grown plants and 24 OTUs in soil-grown plants, it was easy for one or two OTUs to appear very abundant. In our experiment, uncommon and very abundant fungi (over 50%) in sterile sand-grown shoots included *Alternaria, Papiliotrema*, *Penicillium, Phoma, Curvularia* and *Punctulariopsis*; however, when grown on soil, the most abundant fungi changed to *Sarocladium* (FungOTU9) in maize with 99% of the reads, *Penicillium* (FungOTU5) in pea with 99% of the reads, *Penicillium* (FungOTU5) in pea with 99% of the reads, *Penicillium* (FungOTU19) in *Phaseolus* with 78% of the reads, *Sakaguchia* (FungOTU80) in rice with 95% of the reads and *Curvularia* (FungOTU338) in sunflower with 69% of the reads. FungOTU5, 9 and 338 were seed-transmitted since they also occurred in sterile sand-grown shoots, suggesting that soil enhanced their abundance rather than providing starting inoculum. FungOTU19 and 80 (*Sakaguchia*) seem like they may have been deposited in shoots by chance as infrequent inoculum in either seeds or soil. *Sakaguchia* is a Basidiomycete yeast that has previously been observed in the phyllosphere of turfgrasses [105] and rice [106].

## 5. Conclusions

As a complement to a previous study searching for core microbiomes [20], this experiment aimed to document diversity of uncommon bacteria and fungi associated with a panel of 17 important plant species grown inside hermetically sealed jars. Seeds and spermospheres contained some uncommon/abundant bacteria and many fungi, suggesting at least some of the rare microbiome is vertically transmitted. About 95% and 86% of fungal and bacterial diversity inside plants was uncommon; however, by abundance these fungi represent only up to about half of the mycobiome, while less than 11% of bacterial endophytes are rare. When grown on sterile sand, all plants developed microbiomes with uncommon Proteobacteria, Firmicutes, Bacteriodetes, Ascomycetes and Basidiomycetes, proving that seeds can transmit uncommon and varying communities of microbes to the resulting seedlings. A minority of these uncommon vertically transmitted microbes were robust colonizers of soil-grown plants and least of all in rhizospheres, although roots hosted relatively more seed-transmitted microbes, and shoots a greater number still. Except for bacteria inhabiting rhizospheres, soil served as a more diverse source of uncommon microbes than seeds, replacing or excluding the majority of the seed-transmitted microbiome. It was difficult to know whether a microbe was absent from a particular sample because of biotic filtering from the plant, or because of uneven/stochastic inoculum distribution in seeds or soil. By focusing on uncommon microbiomes, a few interesting plant–microbe associations were observed, such as seed transmission and robust endophytic colonization of shoots, roots and/or rhizospheres by the beneficial maize endophyte *Sarocladium zeae*, the phosphate-solubilizing and growth-promoting *Penicillium* in pea and *Phaseolus*, and the stress-resistance-enhancing endophyte *Curvularia* in sugarcane. There was evidence for robust soil transmission into *Arabidopsis* and *Panicum* roots of the phytohormone-producing cyanobacteria *Leptolyngbya*, while the colonization of cassava roots by cassava field-soil-dwelling *Streptomyces* invites speculation that this was a specialized endosymbiont allowed in by its preferred plant host. Some abundant microbes such as *Sakaguchia* in rice shoots or *Vermispora* in *Arabidopsis* roots appeared in no other samples, suggesting that they were stochastically deposited propagules from either soil or seed, making it impossible to know what their relationship may be with other plants. Future experiments that succeed in culturing some of these uncommon microbes would allow for cross-inoculation with high densities of propagules to better understand their host specificity. Cross-inoculation experiments would further help explain microbe role in plant health and productivity, perhaps leading to their future implementation in crop microbiome engineering and agricultural production enhancement. 

## Figures and Tables

**Figure 1 life-12-01372-f001:**
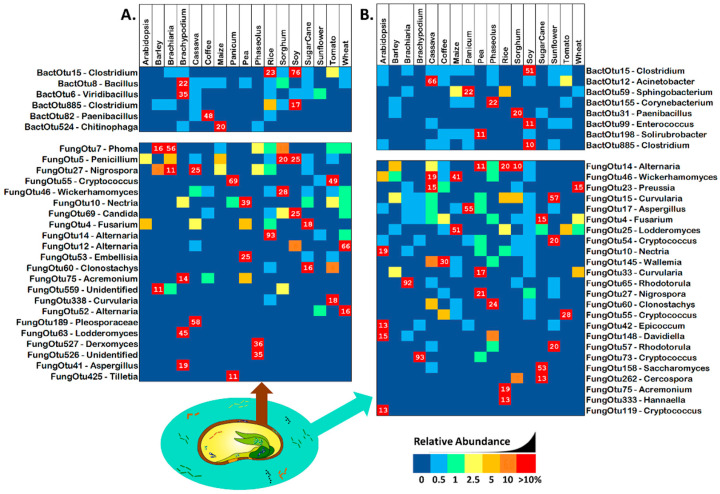
Uncommon and abundant (with a relative proportion of greater than 10% in at least one sample) OTUs of bacterial 16S and fungal ITS in (**A**) seeds and (**B**) spermospheres. Uncommon OTUs were defined as having an occupancy of less than 53% across seeds or spermospheres of the 17 different plant species. Bacterial OTUs are shown in the top blocks, fungal OTUs on the bottom blocks. Next to each OTU ID# is the predicted genus of that sequence. OTU read proportion is represented by color as shown in the legend, with red squares also showing the proportion as a number.

**Figure 2 life-12-01372-f002:**
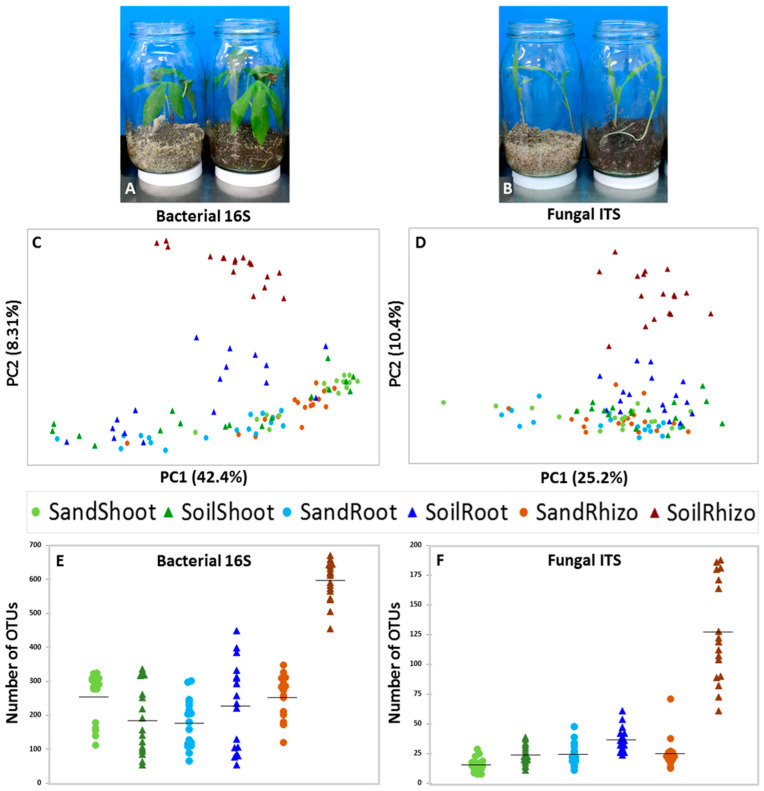
Statistical analysis of total microbial diversity in 17 different plant species grown on sterile sand or field soil. Juvenile cassava (**A**) and sorghum (**B**) plants in sterile sand on the left, field soil mixed with sand on the right. PCA of binary-transformed bacterial 16S (**C**) and fungal ITS (**D**) OTU counts. Scatterplots of the number of different bacterial 16S (**E**) and fungal ITS (**F**) OTUs observed in each sample. Sand-grown samples are displayed as circles, while soil-grown samples are displayed as triangles. Mean values of OTUs observed are indicated by horizontal black bars.

**Figure 3 life-12-01372-f003:**
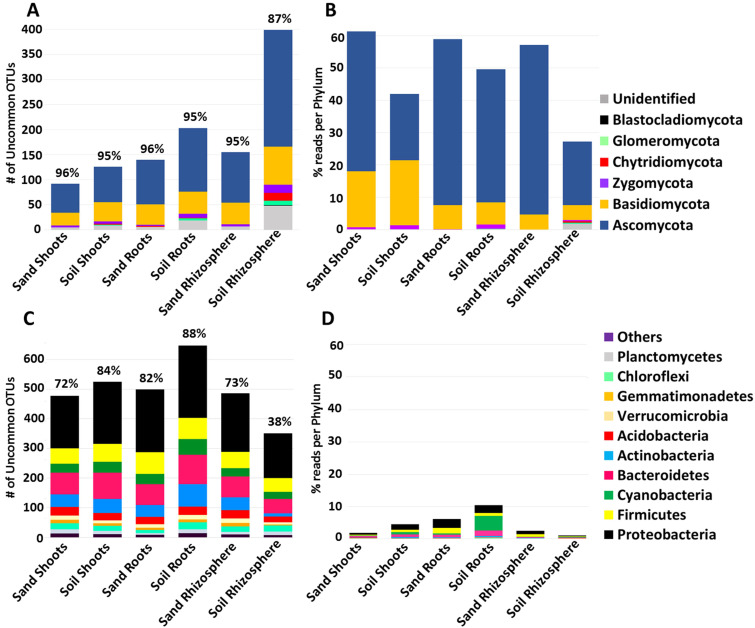
Phylum-level classification of uncommon OTU diversity and read abundance for fungi (**A**,**B**) and bacteria (**C**,**D**) in shoots, roots and rhizospheres. Uncommon OTUs were those that were found in 9 or fewer of the 17 plant species and were calculated by addition across all plant species per sample type. Percentage of uncommon versus total OTUs in each sample type is indicated above each bar. Read proportion per phylum was the result of averaging the number of reads in uncommon OTUs across all plant species. OTU and read taxonomy are indicated by coloring according to the legends at the right.

**Figure 4 life-12-01372-f004:**
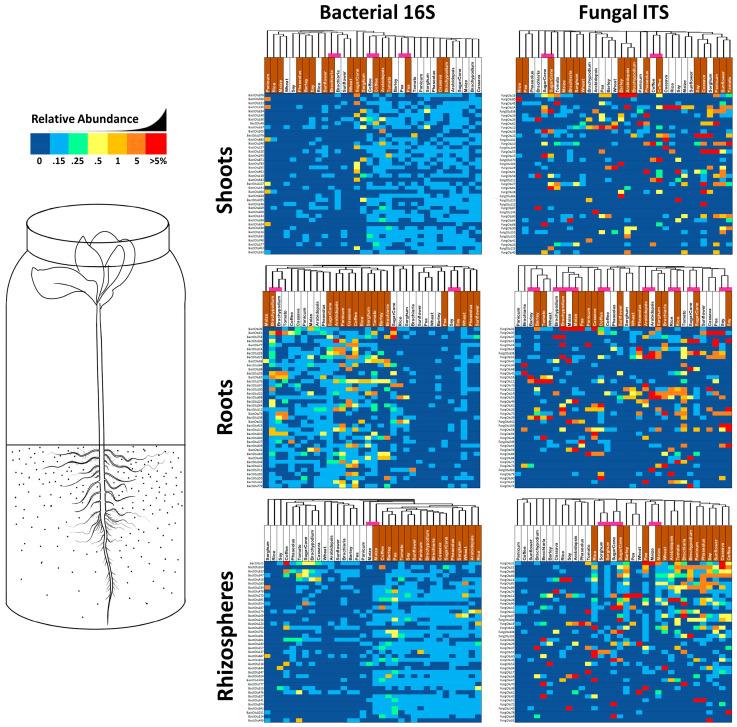
The 40 most abundant yet uncommon OTUs of fungal ITS and bacterial 16S in shoots, roots and rhizospheres of 17 different plant species grown in sealed jars on either field soil or sterile sand. Reads were summed across repetitions and then transformed into relative percentages. Sand-grown plants are labelled in white, while soil-grown samples have a brown label. Samples were organized by Bray–Curtis dissimilarity and those that grouped by plant species are highlighted by a purple block atop the column. Square shading is by percentage value, with dark blue being 0%, up to 0.1% being light blue, between 0.1–0.25% being green, 0.25–0.5% being light yellow, 0.5–1% being dark yellow, 1–5% being orange and greater than 5% being red.

**Figure 5 life-12-01372-f005:**
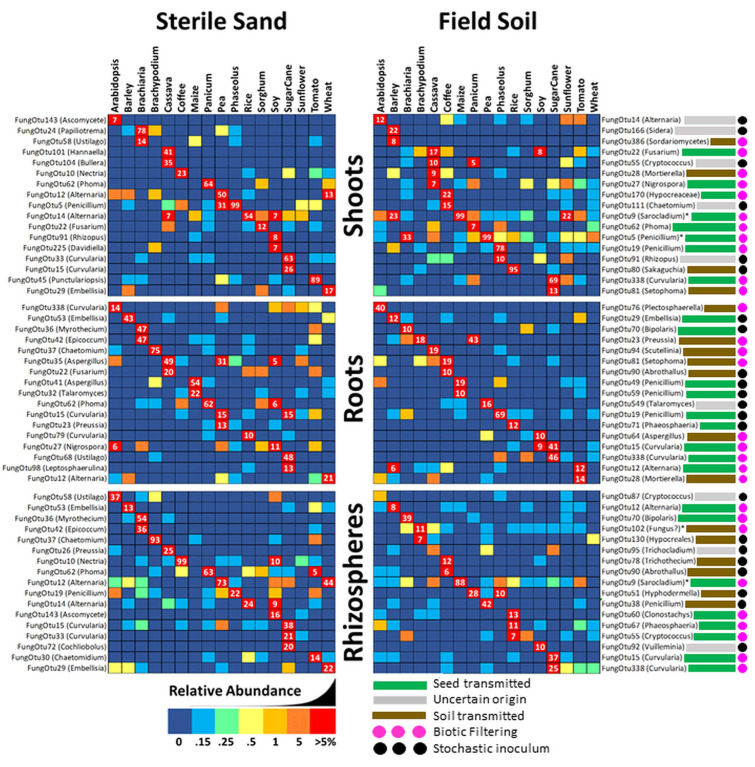
The 17 most abundant fungal ITS OTUs occurring in less than 53% of shoot, root and rhizosphere samples fro m 17 different plant species grown in sealed jars on sterile sand or farm soil. Uncommon reads were added across repetitions and transformed into relative percentages, and the most abundant OTU for each plant is shown. Rows with an asterisk are represented in more than 53% of sampl es; however, they appeared to show a large increase in abundance in one specific plant. Cells are shaded by percentage value, with 0% being dark blue, up to 0.15% being light blue, between 0.1–0.25% being green, 0.25–0.5% being light yellow, 0.5–1% being dark yellow, 1–5% being orange and greater than 5% being red with white numbers inside. Predicted inoculum source in soil-grown plants is shown as gray (unknown), green (seed) or brown (soil) colored bars, while pink dots indicate plants selectively filtering for the fungus and black dots indicate a stochastic supply of inoculum.

**Figure 6 life-12-01372-f006:**
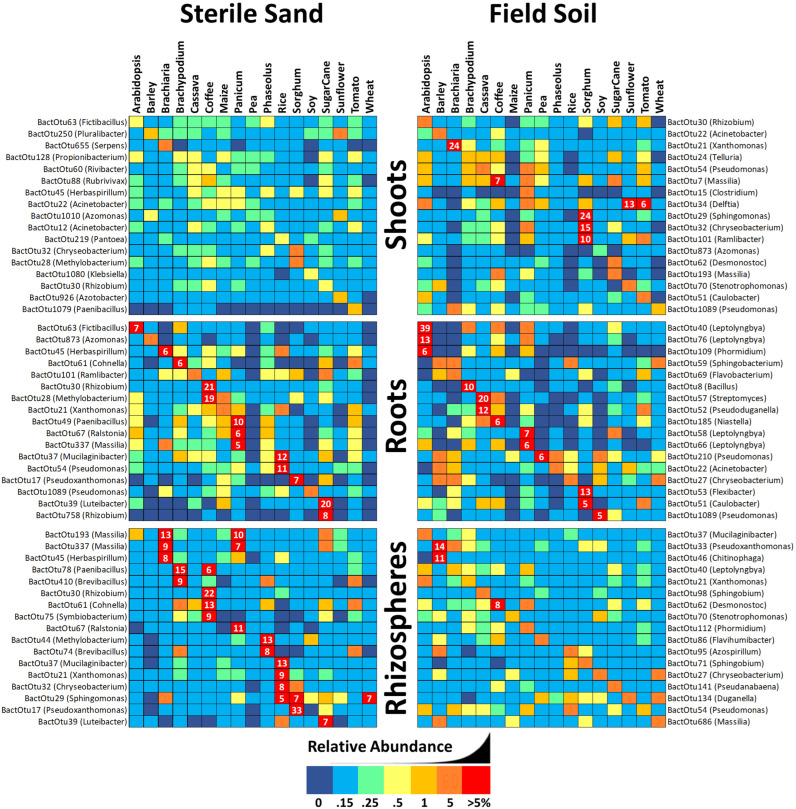
List of the 17 most abundant yet uncommon bacterial 16S OTUs (occurring at a level of more than 0.15% in 9 or fewer different plant species) in shoots, roots and rhizospheres of 17 different plant species raised inside sealed jars on either farm soil or sterile sand. Reads from soil-grown samples were added across repetitions and transformed into relative percentages, and the most abundant OTU for each plant sample is shown. Cells are shaded by percentage value with 0% being dark blue, up to 0.15% being light blue, between 0.1–0.25% being green, 0.25–0.5% being light yellow, 0.5–1% being dark yellow, 1–5% being orange and greater than 5% being red with white numbers inside.

**Table 1 life-12-01372-t001:** Seeds used in this experiment and their provenance.

Species Name	Variety or Genotype	Accession	Provider	Seed Origin
*Arabidopsis thaliana*	Columbia-0		Hazen Lab, U of Massachusetts	USA
*Hordeum vulgare* ssp. *vulgare*	Beaver	CIho 1915	National Plant Germplasm System	USA
*Brachiaria decumbens*	Basilisk	CIAT606	CIAT Genebank	USA
*Brachypodium distachyon*	Bd21		Hazen Lab, U of Massachusetts	USA
*Manihot esculenta*	19	DI-2015	CIAT Genebank	Colombia
*Coffea arabica*	Geisha		Agro Ingenio S. A. S.	Colombia
*Zea mays* ssp. *mays*	B73	PI 550473	National Plant Germplasm System	USA
*Panicum virgatum*	Alamo	PI 422006 01 SD	National Plant Germplasm System	USA
*Pisum sativum*	Aa134	PI 269818	National Plant Germplasm System	USA
*Phaseolus vulgaris*	G19833		CIAT Genebank	Colombia
*Oryza sativa* ssp. *japonica*	Nipponbare	GSOR 100	National Plant Germplasm System	USA
*Sorghum bicolor* ssp. *bicolor*	BTx623	PI 564163 02 SD	National Plant Germplasm System	USA
*Glycine max*	Paramo 29		Semillas del Pacifico S. A. S.	Colombia
*Saccharum officinarum*	CC93-4112 x CC91-1987	CS#725	Cenicaña	Colombia
*Helianthus annuus*	Arrowhead	PI 650649	National Plant Germplasm System	USA
*Solanum lycopersicum*	Heinz 1706	LA4345	C.M. Rick Tomato Genetics Center	USA
*Triticum aestivum*	Prospect	PI 491568 TR04ID	National Plant Germplasm System	USA

**Table 2 life-12-01372-t002:** Sørenson similarity index comparing uncommon OTUs in soil- vs. sand-grown plants.

	Fungal Similarity between Plants			Bacterial Similarity between Plants		
	Rhizosphere	Roots	Shoots	Rhizosphere	Roots	Shoots
*Arabidopsis*	0.05	0.18	0.08	0.17	0.36	0.50
Barley	0.11	0.18	0.30	0.04	0.11	0.30
*Brachiaria*	0.08	0.15	0.11	0.21	0.07	0.07
*Brachypodium*	0.05	0.11	0.00	0.18	0.32	0.47
Cassava	0.06	0.13	0.16	0.14	0.30	0.41
Coffee	0.07	0.17	0.26	0.13	0.31	0.41
Maize	0.09	0.40	0.32	0.12	0.08	0.05
*Panicum*	0.06	0.07	0.19	0.28	0.33	0.11
Pea	0.03	0.15	0.31	0.06	0.17	0.49
*Phaseolus*	0.02	0.24	0.24	0.25	0.17	0.21
Rice	0.34	0.50	0.12	0.10	0.25	0.17
Sorghum	0.10	0.22	0.10	0.13	0.23	0.14
Soy	0.01	0.51	0.48	0.10	0.16	0.11
Sugarcane	0.18	0.33	0.31	0.23	0.39	0.17
Sunflower	0.01	0.13	0.06	0.11	0.30	0.25
Tomato	0.09	0.17	0.38	0.27	0.36	0.46
Wheat	0.02	0.09	0.14	0.10	0.11	0.15
**Average**	**0.08**	**0.22**	**0.21**	**0.15**	**0.24**	**0.26**

## Data Availability

The bacterial and fungal sequence data generated in this study using MiSeq have been deposited and are available in the NCBI Sequence Read Archive (SRA) under BioProject PRJNA731997 and are also provided as Appendix A (annotated sequences and OTU counts) in this publication.

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
