# Peer review of "Stochastic Inoculum, Biotic Filtering and Species-Specific Seed Transmission Shape the Rare Microbiome of Plants"

_life, 2022, doi:10.3390/life12091372_

Round 1

Reviewer 1 Report

It's a great research work, in the future it will be very interesting to be able to grow in the lab these rare microbes and determine their functionality for plants.

I have some questions for the authors:

Why the seed's microbiome were not analyzed before germination or in shorter stages to be able to compare it with the results that are being presented?

From both an ecological perspective and application to agricultural systems, what would be the functionality of that rare microbiome?

It is possible to suggest if rare microbiomes are part of a microbial network and to which "hub" they would belong?

How close they are to being considered that species as a keystone for the plants?

Author Response

Thank you all for your comments and suggestions. Here are my responses to your comments.

Reviewer 1

I will put the reviewer's words in quotations + italics , then write my answer after each question or comment. Also, since there was no explicit request to add to or change anything in the paper, I have not made any changes to the manuscript based on these comments except adding another figure with seed data (now inserted there as figure 1) and modifying the text to reflect the additional information.

"It's a great research work, in the future it will be very interesting to be able to grow in the lab these rare microbes and determine their functionality for plants."

Thank you very much reviewer, I'm glad you liked the work - I spent a lot of time during the pandemic thinking about and writing it. ;-)

"I have some questions for the authors:

Why the seed's microbiome were not analyzed before germination or in shorter stages to be able to compare it with the results that are being presented?"

Seed and spermosphere data was collected for all plants before germination, however it had not been mentioned in the manuscript because it was already a somewhat lengthy paper and because all microbes found in sterile sand grown plants were already known to be seed transmitted, making it potentially redundant to check seed data separately. At your implied request however, I have made up a new figure showing the uncommon taxa in seeds and spermospheres and included the raw data into supplemental table S6. Figure 1 shows rare seed microbes, which helps present the idea that different seeds contained stochastically distributed microbial propagules, thus I think it is a valuable addition to the work.

"From both an ecological perspective and application to agricultural systems, what would be the functionality of that rare microbiome?"

The rare microbiome as we've defined it are microbes that are not found in most plant species. As such, these could either be specialized pathogens or beneficials that can only colonize certain plant hosts (ie. Bradyrhizobium in soybean) or they could be generalist pathogens or beneficials that are restricted in their distribution by limited or random inoculum. In the case of a specialized endosymbiont, identification of its importance might allow plant breeders to select for plants that maintain or enhance the relationship, or it might be possible to search for better microbial partners that can result in better plant phenotypes (eg. searching for super strains of Bradyrhizobium for soybean inoculation). In the case of generalistic beneficials that are limited by inoculum, there is first a need to supply different plants with the microbe to see if the microbe is indeed able to colonize the plant, and if so, there would be a clear opportunity to supply this at high/consistent inoculum densities across the plant population. Inoculating plants with rare propagules at high densities would allow all plants in a farmer's field to enjoy the relationship - out in nature, the lack of this consistent propagule availability may help explain why plants perform differently within an otherwise uniform habitat.  

"It is possible to suggest if rare microbiomes are part of a microbial network and to which "hub" they would belong?"

We've never performed network analysis, but it would be possible. On the other hand, I would venture that we've already got data similar to network analysis in the heatmaps sorted by Bray-Curtis dissimilarity; if there were strong networks of seed transmitted rare microbes, this would cluster samples together by plant type as occurred for the fungi in roots of 9 plant species, while if there were strong soil transmitted networks of rare microbes, this would cluster samples together by soil treatment as occurred for the bacteria in rhizospheres of all 17 plant species. If you'd like us to try include network analysis in the paper, please let us know. 

"How close they are to being considered that species as a keystone for the plants?"

The answer to this question depends on microbe in question. Some such as Sarocladium zeae in maize have previously been identified as beneficial, seed transmitted endophytes. The majority of microbes observed in this study however, have not been studied specifically enough to know how important they might be to plant health and performance, let alone knowing whether they are part of a microbial keystone. Whats more, some rare endophytic microbes may be responsable for redundant plant phenotypes that might be covered by part of an omnipresent core; for example all seeds and plant tissues grown in all conditions contained Pantoea and Enterobacter, thus if a rare microbe is able to help the plant in the same way that Pantoea and Enterobacter do, that phenotype would appear invisible. Nobody has ever generated a microbe free plant which would allow symbiotic hypotheses to be properly tested.

Reviewer 2 Report

The manuscript by Monje et al. investigated microbiomes (including bacteria and fungi) of the shoots, roots and rhizospheres of the 17 most important species of plant (eg. Arabidopsis, Brachypodium, maize, wheat, sugarcane, rice, tomato, coffee, common bean, cassava, soybean, switchgrass, sunflower, Brachiaria, barley, sorghum, and pea). By focusing on these uncommon microbes, significant numbers of rare or uncommon microbes may also play an important role in the health and productivity of certain plants in certain environments. Overall, I like the concept of this study and the methods are technically sound. I would like to point out several issues.

Major comments:

The definition of rare or uncommon microbes in this paper is worth discussing. Whether the same microbe, especially for the same species, can be eliminated before counting. Different plants can recruit different kinds of microbes, but their functions of the microbes may be similar, especially between the same species, and even the genomic differences are very small.The contents of results and discussion are too more to get the points. It is recommended to do the appropriate simplification and content focus. In addition, there are some errors in the writing. Suggest the authors to improve the quality of the content.

Minor comments:

Line 16. must've- > may be.

Line 52-53 the Keywords is too many. Please reduce it to no more than 5.

Line 161-190. -> I suggest that the information about the sources of seed and substrate should be presented in a table, so that it is clearer.

Line 181-182. -> Is there a reference for the type of sterilization (20 min at 121°C)? Or did you have tests to verify the effectiveness of sterilization. What I'm wondering is why not sterilize it with 60Co γ rays (75kGy).

Line 188. -> the glass jars that were 7 cm diameter, instead of wide.

Line 191-207 -> I suggest that the experimental design could be added a flow chart to be more intuitive. In addition, the experiment was not specific enough. For example, in Line 193 and 196, 50% of the soaked seeds were grown in sterile petri dishes, and the other half of the seeds were grown in soil. But there was no follow-up about what they were for?

Author Response

I will put the reviewer's words in quotations + italics , then write my answer after each question or comment. 

Reviewer 2

The manuscript by Monje et al. investigated microbiomes (including bacteria and fungi) of the shoots, roots and rhizospheres of the 17 most important species of plant (eg. Arabidopsis, Brachypodium, maize, wheat, sugarcane, rice, tomato, coffee, common bean, cassava, soybean, switchgrass, sunflower, Brachiaria, barley, sorghum, and pea). By focusing on these uncommon microbes, significant numbers of rare or uncommon microbes may also play an important role in the health and productivity of certain plants in certain environments. Overall, I like the concept of this study and the methods are technically sound. I would like to point out several issues.

Thank you very much reviewer, I'm glad you liked the work - I spent a lot of time during the pandemic thinking about and writing it. ;-)

 Major comments:

The definition of rare or uncommon microbes in this paper is worth discussing.

I understood this to mean that you want me to dedicate more space in the discussion to the definition of rare or uncommon microbes. I reworked the first paragraph of the discussion to focus more on this topic and I think it does a better job of defining rare/uncommon microbes; I hope that you agree.

Whether the same microbe, especially for the same species, can be eliminated before counting.

Hmm, I’m not sure what you mean by this. Perhaps you’re referring to my idea of comparing soil and sand grown plants and subtracting sand grown microbiomes as a way to show a microbe was soil transmitted?  Or perhaps you mean that two different OTUs could be added together based on their taxonomic assignment rather than bioinformatics separation? If the latter, then it is not possible to do since that would change the statistics throughout the whole paper – perhaps you’d like a figure summarizing microbial diversity by genus?

Different plants can recruit different kinds of microbes, but their functions of the microbes may be similar, especially between the same species, and even the genomic differences are very small.

I agree and appreciate this comment. Taxonomy of a bacteria says nothing about its function – a taxonomic group of microbes likely contains both beneficials and pathogens, but there is no way to know based solely on this data. Would you like me to write more about this point in the discussion? 

The contents of results and discussion are too more to get the points. It is recommended to do the appropriate simplification and content focus.

The problem with microbiome studies is that they generate huge amounts of data and papers often have to focus on restricted amounts of the information in order to build a coherent story. As in a previous publication we threw out the uncommon microbiomes, here I wanted to focus on those and you’re right that there is a lot of diverse information which can be hard to wrap our heads around. I cannot shrink down the results much without throwing out data, but I have parsed the discussion and attempted to focus it more on the most important points as you’ve suggested. I hope you agree that it reads better.

In addition, there are some errors in the writing. Suggest the authors to improve the quality of the content.

I have proofread the document again and tried to catch as many other typos, grammatical errors and poorly worded sentences, thank you for your suggestion.

Minor comments:

Line 16. must've- > may be.

Taken care of, since the editor told me to delete the simple summary.

Line 52-53 the Keywords is too many. Please reduce it to no more than 5.

Life journal allows me up to 10. As per your request, I reduced the number of key words, albeit only to 8.  

Line 161-190. -> I suggest that the information about the sources of seed and substrate should be presented in a table, so that it is clearer.

Great suggestion! I’ve done as you requested and embedded the information in a table. Here it is:

Line 181-182. -> Is there a reference for the type of sterilization (20 min at 121°C)? Or did you have tests to verify the effectiveness of sterilization. What I'm wondering is why not sterilize it with 60Co γ rays (75kGy).

Thanks for asking – I did not follow any reference here. To help the methods be more clear, I changed that paragraph to, “Autoclaved glass jars that were 7 cm in diameter and 13 cm tall were filled with 100 mL of twice autoclaved (121°C for 20 minutes) sterile sand before a third autoclave treatment, or they were filled with 100 mL of 1:1 field soil:sterile sand. After planting, 10 mL of sterile distilled water was poured in and the jars sealed with a plastic lid.”. The reason for not using gamma irradiation (which I have used before in the USA and liked for reasons of not making soil chemistry a variable) is that there are no nuclear reactors in the country of Colombia, making it impractical if not impossible for me to have availed myself of that technology. I did not explicitly verify that the sand was sterile, but I hope you’ll agree that three rounds of autoclaving are enough to kill microbes. To help assuage your concerns, I did include conduct sterile sand PCRs to see if there was detectable DNA present (if not living microbes) and was not able to get amplicons after 50 cycles of PCR. In case you missed it, I had already included a sentence mentioning the negative controls, “(note: except for negative controls which did not amplify, reactions were repeated until there was sufficient PCR product for mixing of equimolar amounts of all samples in a 96 plate).”

Line 188. -> the glass jars that were 7 cm diameter, instead of wide.

 Thanks for the suggestion, I changed that.

Line 191-207 -> I suggest that the experimental design could be added a flow chart to be more intuitive.

While I would normally appreciate this idea as well, but the experimental setup was much too simple to necessitate a flow chart that would simply show seeds being germinated in Petri dishes, transplanted to jars (photos or diagrams of these plants already exist in Figure 2 and Figure 4) and then DNA extracted for PCR and sequencing. Sensing that perhaps you didn’t like the relevant methods section, I modified it for clarity and have pasted it below for you to see,

“Of each plant species, either 0.5 g of small seeds or 20 large seeds were soaked for 6 hours in sterile, distilled water within 2 mL or 15 mL tubes. Soaked seed were then transferred to two sterile Petri dishes containing sterile Whatman #1 filter paper (GE HealthCare: USA); seeds in one Petri dish received 3 mL of sterile water, while the other Petri dish received 1 g of field soil resuspended in 3 mL of sterile water. Plates were sealed and incubated at 32°C for several days in the dark until seeds germinated.

Once germinated, sterile grown seedlings were transplanted 2 at a time to glass jars filled with sterile sand, while seedlings germinated in soil were transplanted to jars containing a blend of soil and sand. Jars were incubated in a Panasonic MLR-352H Plant Growth Chamber set at 28°C for 12 hours with 5 lumens of fluorescent light, and for 12 hours at 22°C of dark. Plants were allowed to grow from 2 weeks to 2 months, until they achieved a significant size or until they hit the lid of the jar. Before plants were harvested, lids were detached inside a laminar flow hood, and plants permitted to dry off for 24 hours (Figure 2A, 1B). There were 6 unplanted control jars that were watered with 10 mL of sterile water and incubated in the growth cabinet for 14 days: 3 filled with sterile sand, and 3 filled with a mix of sand and field soil.”

 In addition, the experiment was not specific enough. For example, in Line 193 and 196, 50% of the soaked seeds were grown in sterile petri dishes, and the other half of the seeds were grown in soil. But there was no follow-up about what they were for?

See the above rewritten methods section (2.3). I’m sorry if you were confused by the methods, and I hope that you can better understand what was done now that that part was rewritten.

Round 2

Reviewer 2 Report

The author's reply is very good, I have no opinion on it.